# Do organisms need an impact factor? Citations of key biological resources including model organisms reveal usage patterns and impact

Agata Piekniewska[1], Martijn Roelandse[1,2], Kevin C. Kent Lloyd[3], Ian Korf[4], Stephen Randal Voss[5], Giovanni de Castro[6], Diogo M. Magnani[6], Zoltan Varga[7], Christina James-Zorn[8], Marko Horb[9], Jeffery S. Grethe[10], Anita Bandrowski[1,10]*

1 SciCrunch Inc., San Diego, California, United States of America, 2 martijnroelandse.dev, Ouderkerk aan de Amstel, The Netherlands, 3 Department of Surgery, School of Medicine, Mouse Biology Program, Comprehensive Cancer Center, University of California Davis, Davis, California, United States of America, 4 Department of Molecular and Cellular Biology, UC Davis Genome Center, University of California Davis, Davis, California, United States of America, 5 Ambystoma Genetic Stock Center, Spinal Cord and Brain Injury Research Center, University of Kentucky, Lexington, Kentucky, United States of America, 6 Department of Medicine, UMass Chan Medical School, Worcester, Massachusetts, United States of America, 7 Zebrafish International Resource Center, Institute of Neuroscience, University of Oregon, Eugene, Oregon, United States of America, 8 Division of Developmental Biology, Cincinnati Children's Research Foundation, Cincinnati, Ohio, United States of America, 9 National Xenopus Resource, Eugene Bell Center for Regenerative Biology and Tissue Engineering, Marine Biological Laboratory, Falmouth, Massachusetts, United States of America, 10 Department of Neuroscience, School of Medicine, University of California at San Diego, San Diego, California, United States of America

* abandrowski@ucsd.edu

## Abstract

Research resources like transgenic animals and antibodies are the workhorses of biomedicine, enabling investigators to relatively easily study specific disease conditions. As key biological resources, transgenic animals and antibodies are often validated, maintained, and distributed from university-based stock centers. As these centers heavily rely on grant funding, it is critical that they are cited by investigators so that usage can be tracked. However, unlike systems for tracking the impact of papers, the conventions and systems for tracking key resource usage and impact lag. Previous studies have shown that about 50% of the resources are not findable, making the studies they support irreproducible, but also makes tracking resources difficult. The RRID (Research Resource Identifiers) project is filling this gap by working with journals and resource providers to improve citation practices and to track the usage of these key resources. Here, we reviewed 10 years of citation practices for five university based stock centers, characterizing each reference into two broad categories: findable (authors could use the RRID, stock number, or full name) and not findable (authors could use a nickname or a common name that is not unique to the resource). The data revealed that when stock centers asked their communities to cite resources by RRID, in addition to helping stock centers more easily track resource usage by increasing the number of RRID papers, authors shifted from citing

**Data availability statement:** Data underlying figures are available within the paper and its Supporting Information files. Raw data is available via the SciCrunch.org web portal (e.g., https://rrid.site/data/record/nlx_144509-1/SCR_002953/resolver/mentions?q=MMRRC&i=rrid:scr_002953), and is also accessible via several resolver services including (https://n2t.net/RRID:SCR_002953), and a twitter as well as a new BlueSky bot (https://bsky.app/profile/rridrobot.bsky.social) and via an open API (elastic search).

**Funding:** U24DK097771, R24GM144308, 75N97021P00135, U24AI126683, P40OD028116

**Competing interests:** COI: AP is an analyst at SciCrunch Inc COI: MR serves as an independent contractor for SciCrunch. COI: Part of this work was supported by the University of California, San Diego, Center for Research in Biological Systems and grants from NIH (Award #U24DK097771). Dr. Grethe has an equity interest in SciCrunch, Inc., a company that may potentially benet from the research results. The terms of this arrangement have been reviewed and approved by the University of California, San Diego in accordance with its conict of interest policies. COI: Dr. Bandrowski is a co-founder and current CEO of SciCrunch, Inc., a company that may potentially benet from the research results. The terms of this arrangement have been reviewed and approved by the University of California, San Diego in accordance with its conict of interest policies.

**Abbreviations:** AGSC, Ambystoma Genetic Stock Center (RRID:SCR_006372); DOI, Digital Object Identifier; EXRC, European Xenopus Resource Center (RRID:SCR_007164); MMRRC, Mutant Mouse Resource and Research Center (RRID:SCR_002953); MOD, Model organism database (the genomic authority for the animal); NIH, National Institutes of Health; NHPRR, Nonhuman Primate Reagent Resource (RRID:SCR_012986); NXR, National Xenopus Resource (RRID:SCR_013731); PID, Persistent Identifier; PMID, PubMed Identifier (RRID:SCR_004846); PMCID, PubMed Central Identifier (RRID:SCR_004166); RRID, Research Resource Identifier (RRID:SCR_004098); ZIRC, Zebrafish International Resource Center (RRID:SCR_005065); DSHB, Developmental Studies Hybridoma Bank (RRID:SCR_013527)

resources predominantly by nickname (~50% of the time) to citing them by one of the findable categories (~85%) in a matter of several years. In the case of one stock center, the MMRRC, the improvement in findability is also associated with improvements in the adherence to NIH rigor criteria, as determined by a significant increase in the Rigor and Transparency Index for studies using MMRRC mice. From these data, it was not possible to determine whether outreach to authors or changes to stock center websites drove better citation practices, but findability of research resources and rigor adherence were improved.

## Introduction

Stock centers serve as both archives and distribution centers of research animals, materials, data, and information. Stock centers ensure resource availability and accessibility to the scientific community. These centers typically confirm identity, verify genetics, maintain viability, and certify the health status of their holdings. In one case, the AGSC stock center maintains one of the last large colonies of *Ambystoma mexicanum*, a species that once thrived in aquatic habitats that are now dwindling near present-day Mexico City. Because there may not always be a good profit margin in selling biologically important research animals, stock centers are typically funded by grants while recovering some of their operating costs by distribution fees. Resource centers such as the Nonhuman Primate Reagent Resource (NHPRR) or the Developmental Studies Hybridoma Bank (DSHB), which provide antibodies, are also primarily grant-funded and are therefore under similar pressures. Unlike commercial firms, which maintain only commercially viable common stocks, the mechanism of funding for stock and resource centers necessitates that they track the biological and translational impact to maintain funding, like other grantees. Research resources like these are the workhorses of biomedicine, enabling investigators to study specific disease conditions (see Kiani et al 2022 [1]; Bergen et al, 2022 [2]; Garcia-Garcia, 2020 [3]). However, while grantees generally can use citation indexes to track impact, stock centers have to manually track the use of their research resources.

The purpose of RRIDs (Research Resource Identifiers) is primarily to serve investigators to ensure that they, or their readers, will easily find the exact research resources used in a study (Bandrowski et al, 2015 [4]; Marcus et al, 2016 [5]; Bandrowski, 2022 [6]). For rigor and transparency reporting standards that cover research resources, such as the ARRIVE, MDAR, and JATs, have also incorporated RRIDs as part of their recommendations for better reporting practices for research resources (Percie du Sert et al, 2020 [7]; Macleod et al, 2021 [8]; NISO JATS Ver 1.2 [9]). However, for resource providers, these identifiers are also a means to track the usage of resources throughout the scientific literature. Indeed, the role of SciCrunch is to both encourage the use of RRIDs and, via a contract from NIH, to track the usage of research resources. Stock centers have promoted the use of RRIDs to varying degrees starting around 2016–2020 through listing them on their websites and encouraging good citation practices within their communities. Most stock centers

encourage authors to use RRIDs in more than one way, and there are often sustained campaigns over several years encouraging their use in addition to the more visible changes to the stock center website.

In this study, we sought to determine whether efforts of resource centers to promote the use of RRIDs and good citation practices for research resources improve the citation practices and overall transparency of published papers by their respective communities. We measured the citation practices of the research community for five university-based resource providers over 10 years spanning the period before and after RRIDs were introduced and evaluated the ease with which automated routines could identify the organism or reagent used.

## Methods

NIH Office of the Director (OD) funded organism stock centers:

Mutant Mouse Resource and Research Center (MMRRC), National Xenopus Resource (NXR), Ambystoma Genetic Stock Center (AGSC), Zebrafish International Resource Center (ZIRC)

For this study, curators collected publications from Google Scholar that were published between Jan 1, 2011, and Dec 31, 2022, and contained keywords relevant to the OD-funded stock centers NXR, AGSC, or ZIRC. Pubmed Central (PMC) open-access publications were used to collect publications that mentioned MMRRC. PMC was used to search for MMRRC publications because Google Scholar does not offer filtering based on accessibility, and analysis on the total collection of MMRRC papers (4,180) was not feasible. The search criteria used to find publications can be viewed in S1 File. Publications were also collected from SciCrunch's internal database using Metabase (RRID:SCR_001762), which contains a list of RRID citations gathered by curators over the last 8 years into Hypothes.is (RRID:SCR_000430). The two lists were combined, and duplicate papers were removed. Books, PDFs, redacted publications, publications without a methods section, and papers that may have used the species as 'wild caught' or without any discernible reference to the stock center or animal as a stock were excluded from the study. We were unable to determine if stocks were shared between laboratories, how long these animal colonies were maintained, or whether genetic drift was being accounted for in the laboratories because that information is generally not available in manuscripts. The data were collected in March 2023 to provide this citation information to stock centers as part of an NIH-OD contract, thus, the study was not pre-registered. Note that 2022 is a partial year, as some publishers reveal data to PMC with some delay. For this study, a total of 2,662 original research papers were collected. Each paper referenced a particular stock center as follows: MMRRC 1,849, NXR 191, AGSC 180, ZIRC 443.

Curators classified each resource citation based on the context found in the publication. To classify each citation, the curator read relevant sections of each publication and searched for key terms related to each stock or stock center to determine the context. The persistent identifier (PMID, PMCID, DOI) and publication date were collected and stored in Google Sheets.

The two main categories of citations are: "an animal was used" and "an animal was not used". In the second category, the author(s) may have mentioned organizing a workshop, providing care instructions to researchers, provision of data, and general correspondence between the stock center and researchers. For the analysis of identifiability of stocks, we focus on the category of citations where an animal was used, though this is not necessarily the only part of the total activities of the stock center. For citations of animals, the following categories were obtained (Table 1 for example sentences of each category): RRID of the animal is provided in the paper, the catalog number for the animal is provided and RRID determined by curator, the name of the animal stock is provided and RRID determined by curator, nickname of the animal stock is provided but RRID could not be determined and "other" category when wrong RRID was provided, catalog number was provided but RRID couldn't be determined or animal was mentioned but RRID of the stock center was provided. In the "nickname" group, the authors used a name of the stock that was not the official name of the stock or a known unique short name of the stock according to the stock center or MOD at the time that the data were captured. For statistical analysis, these categories that reference an animal were "lumped" into two main sub-categories: findable resources (blue shaded) and not-findable resources (orange shaded).

 

**Table 1. Citation practice category examples from original papers broken down by type.**

| | Example sentence |
|---|---|
| RRID of resource provided | Xenopus were handled following the NIH Guidelines for Use and Care of Laboratory Animals that were approved by the Institutional Animal Care and Use Committee at the Marine Biological Laboratory, Woods Hole, MA. In Fig 1C and D, Xtr.Tg(pax6:GFP;cryga:RFP;act1:RFP)Papal (**RRID:NXR_1021**) embryos were injected with 20 ng of mo-e15i15 or standard mo-control and analyzed at stage 45. |
| Catalog number provided | Experimental models: Organisms/strains: Zebrafish: AB wild-type; both males and females; 3–18 months; 12 hpf (**ZIRC Cat# ZL1**). |
| Full name provided | 5xFAD mice (**B6SJL-Tg(APPSwFlLon,-PSEN1\*M146L\*L286V)6799Vas/Mmjax**) were purchased from The Jackson Laboratory (Bar Harbor, ME, USA) and maintained in accordance with the laboratory guidelines. |
| Nickname provided | Mice with Cre recombinase-ERT2 fusion gene driven by ROSA26 promotor (B6.129-Gt(ROSA) 26Sortm1(cre/ERT2)Tyj/J from The Jackson's Laboratory) and **Spry2f/f** (MMRRC, mutant mice research and resource center) were crossed to generate mice carrying ERT2-Cre:Spry2f/f. -or- The **anti-CD8** antibodies were kindly provided by **Dr. Keith Reimann**. |
| Other | We thank the National Xenopus Resource (**RRID:SCR_013731**) for providing **Nkx-2.5:GFP** transgenic frogs. |

## The OD-funded Nonhuman Primate Reagent Resource (NHPRR)

A total of 545 papers were analyzed for the NHPRR resource. The method of collecting these citations was different from the above stock centers in that the dataset consists of papers collected by the NHPRR team. For this study, all data were combined with the RRID papers already collected by the RRID curators and deduplicated as above. The curator read and categorized each reference in the same manner, with the same categories as above, without further input from the NHPRR staff. The salient difference is that the literature covered should be considered far more complete, and the search strategies included papers that were discovered by NHPRR staff by a myriad of strategies only available to the stock center, such as the names of people who made purchases and cited the resource "as previously described". These references frequently acknowledge the NHPRR by nickname, principal investigator, or fail to acknowledge the NHPRR but use reagents uniquely provided by NHPRR. This type of analysis is far more time-intensive. The curator categorized each citation according to the rubric created below.

## RRID implementation at stock centers

**MMRRC:** Each stock center implemented RRIDs in a slightly different way and at slightly different times, which may lead to different outcomes. The MMRRC implementation began in 2016 with some text supporting RRIDs on January 8th, and an update to the website that authors use to order animals on November 15th of 2016 (Fig 1). This website was changed to contain a "Citation ID" which both shows the RRID and contains a button to help authors copy the citation. The MMRRC also began a campaign of awareness in 2016 for their users, asking them to use RRIDs when citing mice via social media and scientific conferences.

**NXR:** Both Xenbase and NXR worked to gain agreement on RRIDs for frog stocks in the summer of 2017, converting catalog numbers into RRIDs, harmonizing data records in both locations, and then announced the RRID initiative on both Xenbase and NXR websites on December 13th, 2017. Stocks shared with other resources (e.g., EXRC) were given unique RRIDs to help establish provenance going forward. In 2022, the NXR website was updated to include RRIDs directly on frog web pages (see Fig 1).

**AGSC:** In 2017, RRIDs were assigned to all axolotl stocks on the AGSC website. RRIDs were made more visible via a 2019 website update that improved the shopping experience of users. RRIDs are visible to users when non-custom

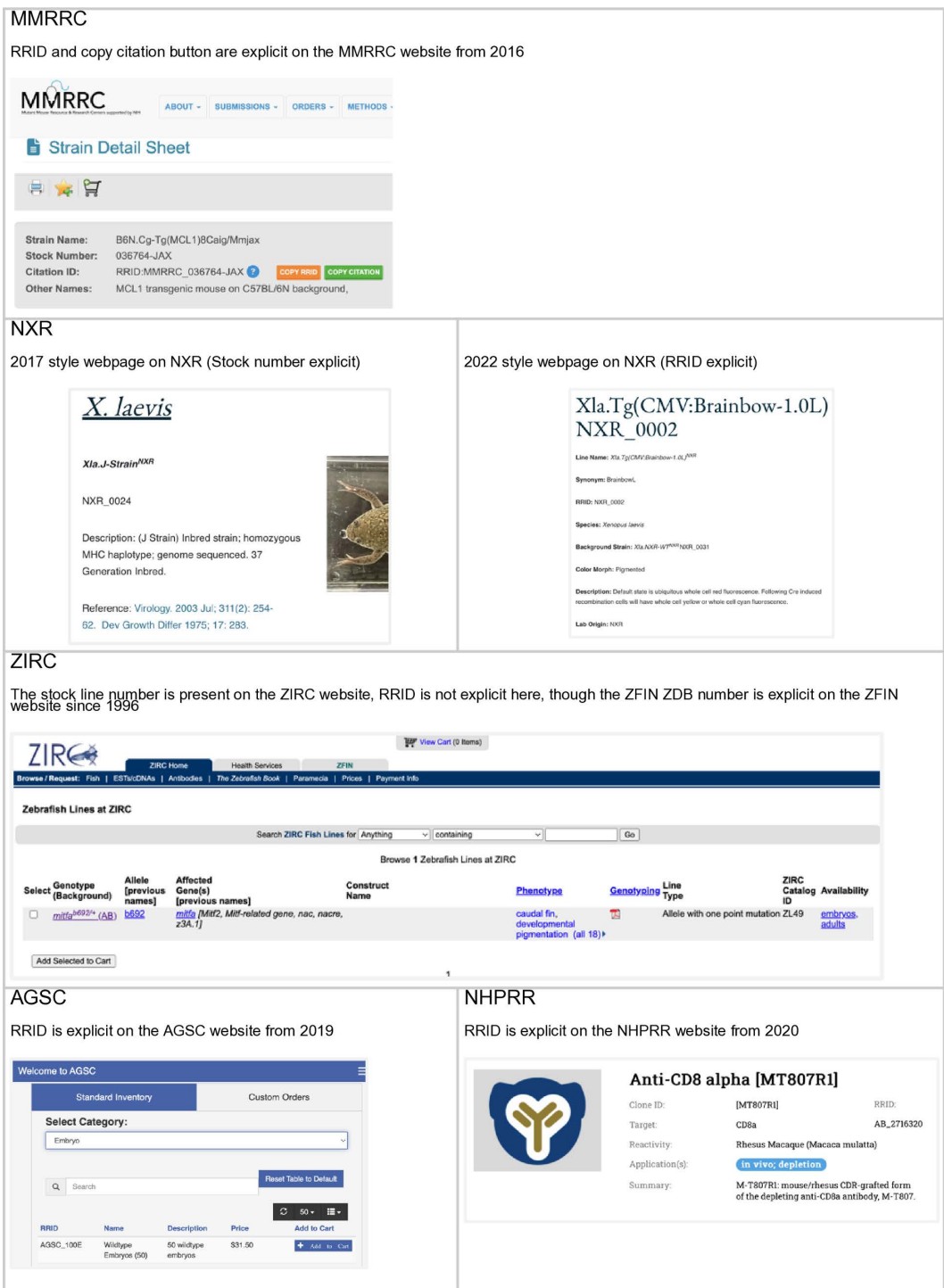

**Fig 1. A summary of a typical stock webpage for each stock center.**

stocks and supplies are selected for purchase. The user community was made aware of axolotl RRIDs in a publication [Voss et al, 2019] and encouraged to use axolotl RRIDs in community newsletters. With new NIH requirements and support for curation and informatics activities, the AGSC and other stock centers will have greater capacity to raise RRID awareness.

**ZIRC:** While ZIRC does not display the RRID syntax on the website, all ZIRC catalog identifiers, which are present on the ZIRC website, can be combined with the prefix "RRID:ZIRC_" to create resolvable RRIDs. The general format is: [RRID:ZIRC_CatalogID]. For example, for fish lines, the format is [RRID:ZIRC_ZL#]. Thus, the correct identifier for the AB wild-type line is RRID:ZIRC_ZL1, whereas the $slc45a2^{b4/+}$ mutant line is identified by RRID:ZIRC_ZL85 (https://zebrafish.org/fish/lineAll.php). All ZIRC entries are also linked to unique ZFIN ZDB identifiers (https://zfin.org/), which encourages the usage of PIDs (persistent identifiers).

**NHPRR**: The NIH Nonhuman Primate Reagent Resource had several reagents added to antibodyregistry.org in October 2017, but the bulk of the resources were added in January 2020. The NHPRR website was redone and updated in August 2020, and was designed to display RRIDs for all products prominently. The RRID is currently listed in product labels, technical datasheets, shipping manifests, invoices, and publication lists. Reagents are searchable by their RRIDs throughout the website. Further, tailored citation guidelines are provided for each product (also accessible by scanning the product's QR code), and a general 'How to Cite our Reagents Using Resource Reagent Identifiers (RRID)' document is sent with each order. Several RRID dissemination campaigns were performed after the website release, including website announcements and three conference presentations for NHP research communities.

There is no true control stock center which never in any way joined the RRID initiative, which is a substantial limitation of this study. Our funding was specifically targeted towards the NIH Office of the Director supported stock centers, all of which were part of the RRID initiative in some way, and the data were collected for those. ZIRC may be the closest to a control because the stock center has not added RRIDs to their website, though these can be easily created from information available on the stock center website. We can find correlations between the timing of events at the stock centers and the scientific literature based on this dataset.

## SciScore

SciScore is a specialized methods review tool for scientific articles. It employs 55 algorithms to evaluate submitted methods sections against a wide range of rigor criteria known to impact the reproducibility of scientific research. In addition to this, SciScore identifies sentences containing key research resources—such as antibodies, cell lines, plasmids, and software tools—and assesses how uniquely identifiable these resources are based on the accompanying metadata. Using this analysis, SciScore generates a reproducibility score and a detailed report, which includes a rigor adherence table and a key resources table.

The curator collected the methods section from each paper and obtained the Reproducibility and Transparency Index RTI (Menke et al, 2020 [10]) for each paper using the SciScore tool (version 2, https://sciscore.com, RRID:SCR_016251). Publications were grouped by stock center and year, and the average SciScore for each year was calculated. These publications were compared with the RTI data published in (Menke et al, 2022 [11], Appendix 5), which is based on scores of 1,813,865 papers (sum of papers 2011–2020). The same major version of the SciScore tool was used in both studies.

## Statistics

Three analyses were conducted after the data were collected. The Z-test for independent proportions was used to evaluate the hypothesis that the proportion of non-findable resources (orange colors in all figures) changed after the RRID inclusion in the literature. The assumption of independence and sample sizes were checked. The statistics were calculated by the online calculator (RRID:SCR_016762) and are provided in the part - "Proportion of 'findable' and 'not-findable' resources" below.

In the second analysis, we used linear regression to estimate values of overall RTI for 2021 and 2022, as the overall analysis published in Menke et al, 2022 [11] is available up to 2020. The assumption of independence of samples, linearity, and normality were checked and not violated.

We also used a two-tailed t-test for independent samples to evaluate the hypothesis that the difference between RTI for MMRRC papers and the overall set after 2016 is significant (2016 marks the implementation of RRID by MMRRC). The normality and homoscedasticity assumptions were not violated. The second and third analyses were performed with Jamovi software (RRID:SCR_016142). Results and details are available in the Analysis of the Rigor and Transparency Index section of Results.

## Results

### Analysis of resources

We report from a total of 3,208 papers referencing resource centers in any context (2,663 of stock center papers and 545 of NHPRR papers). Table 2 provides a summary of the number of citations analyzed for stock centers and NHPRR, broken down by type, including categories when no animal/antibody was used.

For all subsequent figures and analyses, the focus is only on categories where an animal or antibody was used. Fig 2 shows a breakdown of citations to resources or stock centers by type for all citations between 2011 and 2022. Together with Table 2, it shows that in all stock centers, except AGSC, the most common category of citations was a category that we considered "findable" (various shades of blue in all figures), meaning that it is most common to find a findable citation to a stock. The one exception to this is AGSC, which is a small community where the most common way to define a stock is by the nickname, which may be sufficient for investigators within the ambystoma community but may be difficult to identify from outside of the community.

Grouping citations by year reveals the trend over time, in which nearly all centers move toward citation by one of the "findable" (blue) categories. This is perhaps most clear for MMRRC, where before 2014 (the start of RRID pilot project) roughly half of the citations were in the findable category and half were in the "not findable" category, but after 2016 (the point when the MMRRC website was changed to make RRIDs a preferred citation method) authors became far more likely to cite mice using the RRID and far less likely to use a nickname (checked with the Z-test for proportions, $z = -7.1052$, $p < 0.0001$, see details in "Proportion of "findable" and "not-findable" resources" section).

**Table 2. Summary of data.**

| | MMRRC | | NXR | | AGSC | | ZIRC | | NHPRR | |
|---|---|---|---|---|---|---|---|---|---|---|
| | Citations | Papers | Citations | Papers | Citations | Papers | Citations | Papers | Citations | Papers |
| RRID of resource provided | 676 | 511 | 69 | 50 | 39 | 23 | 88 | 66 | 34 | 28 |
| Catalog number provided | 850 | 704 | 0 | 0 | 3 | 2 | 34 | 28 | 12 | 10 |
| Full name provided | 237 | 216 | 2 | 2 | 0 | 0 | 200 | 156 | 330 | 277 |
| Nickname provided | 381 | 353 | 33 | 27 | 132 | 124 | 138 | 109 | 314 | 239 |
| Other | 0 | 0 | 26 | 23 | 1 | 1 | 2 | 2 | 1 | 1 |
| **Total - animal/antibody was used** | **2,144** | **1,761** | **130** | **93** | **175** | **151** | **476** | **330** | **691** | **516** |
| RRID SCR of stock center provided (no animals used) | 14 | 14 | 47 | 47 | 14 | 14 | 1 | 1 | 0 | 0 |
| Stock center mentioned only without RRID (no animals used) | 74 | 74 | 64 | 64 | 29 | 29 | 115 | 116 | 29 | 29 |
| **Total- no animal/antibody was used** | **88** | **88** | **111** | **111** | **43** | **43** | **116** | **117** | **29** | **29** |
| **Total curated** | **2,232** | **1,849** | **241** | **191** | **219** | **180** | **579** | **443** | **720** | **545** |
| **Google Scholar search totals (1/5/2024)** | | **4,200** | | **307** | | **293** | | **527** | | **86** |

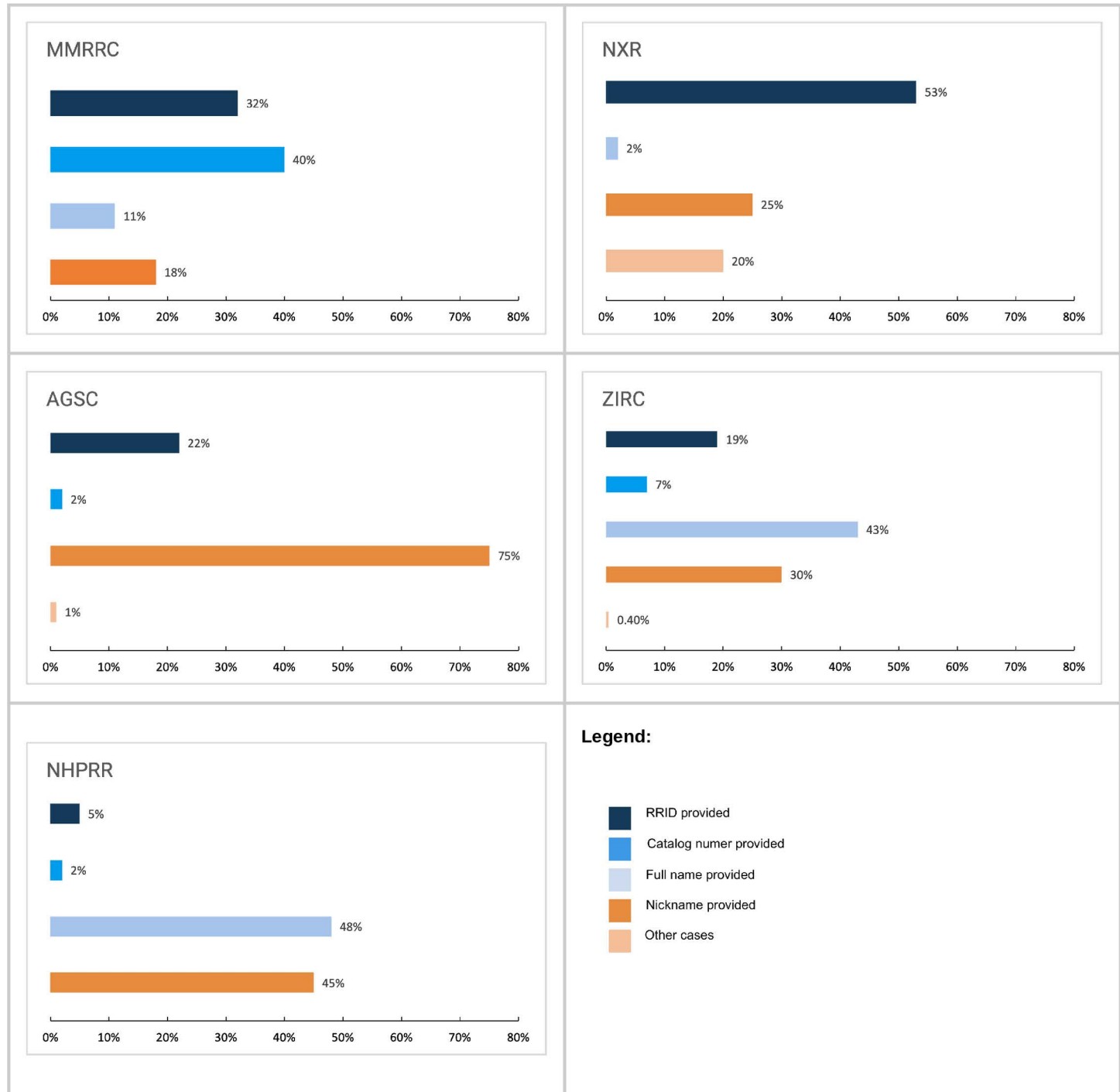

**Fig 2. Breakdown of citations to resources or stock centers by type for all citations between 2011 and 2022.**

Fig 3 shows this trend over time for each stock center, and the most notable shifts in the rate of compliance appear to be the year or two following the stock center adding RRIDs to their website. Interestingly, the NHPRR joined the RRID initiative quite late. In 2020, they registered all of their antibodies and began to display the RRIDs on their new website. This stock center, therefore, can be considered a control since journals were asking for RRIDs since 2014 (for a few journals and 2016 for Cell Press journals), but the NHPRR was not supporting their effort until 2020. If authors wanted to comply with the journal's policy to add RRIDs, they would have had to register the RRIDs for each antibody. Indeed, there were several antibodies registered by diligent authors before 2020, but compliance was minuscule. Interestingly, our data show that authors from 2011 to 2019 continue to exhibit roughly the same rates of using nicknames for NHPRR antibodies. Thus, until the stock center began to increase awareness among NHPRR authors, the rate of non-findable antibodies continued to be roughly 40%. Over a couple of years, the non-findable antibody references may show a trend similar to the other stock centers, however, our analysis of findable to not-findable resources does not reach significance.

The RRID is not highly visible on the ZIRC repository, but the pattern of citation roughly follows the other stock centers, which may reflect the overall trend of increasing usage of persistent identifiers, as those are indeed reflected on the ZFIN authority website in addition to ZIRC.

## Proportion of "findable" and "not-findable" resources

We tested whether the implementation of RRIDs changed the proportion of identifiable resources in the literature. We used the Z-test for proportions to test if the proportion of identifiable resources is significantly different. The assumptions of independence of samples were evaluated and confirmed by the analysis design. As some stock centers don't have many citations in the first years of analysis, we decided to combine numbers for three years at the beginning of the analysis and numbers for the last three years, and compare numbers of findable citations (all blue categories) with not-findable citations (all orange categories). We found that there are significant changes in the proportion of identifiable resources in all centers except NHPRR. This center is the most recent to join, and so the first 3 years of implementation are being counted. As there is a fairly clear trend in Fig 3, with RRIDs increasing each year, and in 2022, the "not-findable" category is beginning to give way to more findable categories, thus we anticipate that this trend will become significant at some future time (Table 3).

## Analysis of the rigor and transparency index

The Rigor and Transparency Index (RTI, version 2.0) is based on SciScore's automatic assessment of the rigor and transparency of papers on criteria found in reproducibility guidelines (e.g., Materials Design, Analysis, and Reporting checklist criteria). The PubMed Central open access papers in biomedicine were all assessed in 2020 (Menke et al, 2022 [11]), and we used the overall numbers of the RTI per year for the currently manually analyzed papers. Unfortunately, the 2021 and 2022 years are not available for this index, therefore, we extrapolated the likely trajectory for 2021 and 2022. We used linear regression because it is relatively unlikely that the overall literature will shift substantially in two years, because significant shifts were only visible in individual journals such as Nature at the time that the checklist was being implemented (see Menke et al, 2020 [10]). In the absence of major editorial policy shifts, the expectation is that the RTI will increase slightly year over year. The regression line we obtained is $\hat{y} = 0.04412X - 84.9523$ ($R^2 = 0.950$, F = 152, p < 0.001), which gives us an estimate for 2021 RTI = 4.21 and 2022 RTI = 4.26. The assumptions of independence of samples, linearity, and normality were not violated.

Like Menke et al (2022 [11]), here we used SciScore (Ver 2, RRID:SCR_016251) to check each paper that contained a reference to an animal or antibody from our exemplar stock centers. Fig 4 shows the comparison of the overall RTI vs the papers gathered that refer to one of the stock centers. With MMRRC, the RTI largely overlaps the MMRRC papers until 2016, where the two lines begin to diverge, suggesting that papers that make use of MMRRC mice are significantly better at following rigor and transparency guidelines than the average paper. We confirmed the hypothesis that the difference

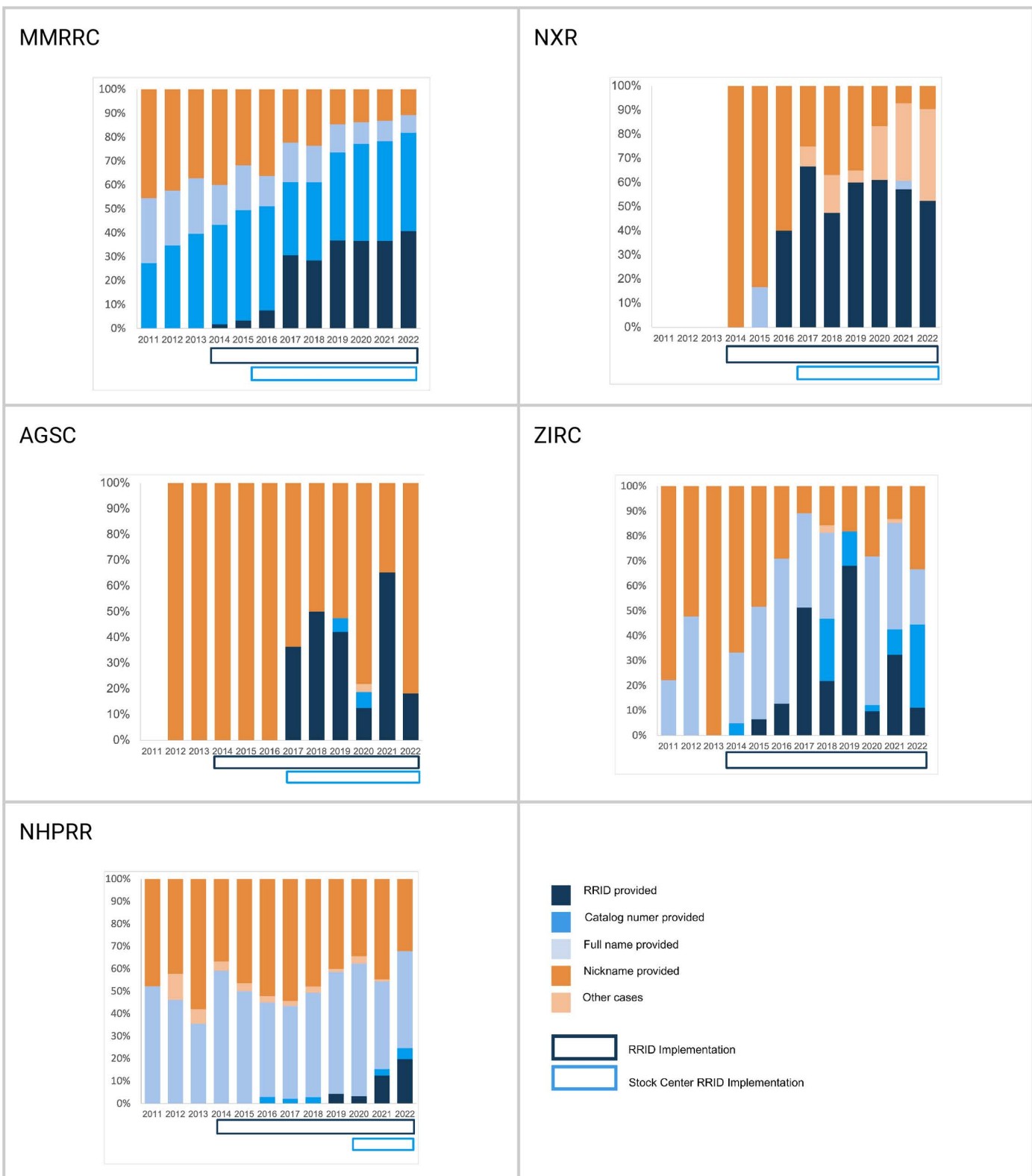

**Fig 3. Citations of research resources over time.** Notable dates: Feb 1, 2014, start of the RRID project (25 neuroscience journals), 2016 eLife and Cell Press join the RRID initiative (Cell Press does so by deploying the STAR tables), 2018 Nature joins the RRID initiative, 2016 MMRRC begins displaying

RRIDs with copy function on website, 2017 NXR begins a campaign to ask the xenopus community to use RRIDs, 2017 AGSC begins outreach campaign within salamander use community, ZIRC, 2020 NHPRR registers all antibodies to obtain RRIDs and begins campaign with users of the resource.

**Table 3. Statistical treatment of resource findability.**

|  | Findable Resource Citations | Not-Findable Resource Citations | Total | Z-statistics and p-value |
|---|---|---|---|---|
| MMRRC |  |  |  |  |
| 2011-2013 | 48 (59%) | 33 | 81 | z=−7.1052, |
| 2020-2022 | 1,160 (88%) | 165 | 1,325 | p<0.0001 |
| NXR |  |  |  |  |
| 2014-2016 | 3 (25%) | 9 | 12 | z=−2.1231, |
| 2020-2022 | 39 (58%) | 28 | 67 | p=0.03375 |
| AGSC |  |  |  |  |
| 2012-2014 | 0 (0%) | 37 | 37 | z=−4.114, |
| 2020-2022 | 23 (35%) | 42 | 65 | p=0.0004 |
| ZIRC |  |  |  |  |
| 2011-2013 | 12 (33%) | 24 | 36 | z=−5.0378, |
| 2020-2022 | 171 (75%) | 57 | 228 | p<0.0001 |
| NHPRR |  |  |  |  |
| 2011-2013 | 39 (49%) | 40 | 79 | z=−1.78083, |
| 2020-2022 | 150 (61%) | 97 | 247 | p=0.07494, NOT significant |

between RTI for MMRRC papers and the overall set after 2016 is significant with a two-tailed t-test for independent samples (t=4.27, df=10, p=0.002, Cohen's effect size 2.5 (large)). According to the Shapiro-Wilk and Levene tests, the normality (W=0.967, p=0.875) and homoscedasticity (F=0.044, p=0.837) assumptions were not violated.

The papers using animals from the other stock centers do not show any trend in improving rigor criteria adherence, but NHPRR may be demonstrating an early trend in the same direction as MMRRC, though the trend is not significant. Of course, one of the features measured within the RTI is the findability of research resources therefore, it may be somewhat unsurprising that these scores begin to diverge. If, however, the major difference was simply the presence of more findable resources over time, then papers from all stock centers except NHPRR would be predicted to also shift in line with their resource findability. However, the other stock centers do not show any shifts away from the mean, suggesting that something about the recent papers using MMRRC and potentially NHPRR may be shifting authors toward more transparent and rigorous practices compared to the average paper. It seems more likely that author outreach activities from the stock centers may be impacting the transparency of published papers.

## Discussion and conclusions

In this study, we examined how authors are citing research resources that are housed at five university-based stock centers to determine if implementing RRIDs correlates with changes in citation practices and other rigorous science practices. The data show clearly that RRID implementation correlates with changes in citation practice, because citation practices go from about half not findable to mostly findable in nearly all stock centers, just as RRIDs are implemented at the stock center.

The trend seen for nearly all stock centers reveals that the RRID project itself, with only journals enforcing author behavior, is less successful compared to the journals and stock centers attempting to change author behavior. While there are >1000 journals allowing RRIDs to be published, there are only about 100 that actively enforce them (see

**Fig 4. The Rigor and Transparency Index, a comparison of all papers published in a given year [Menke 2022] vs the papers that refer to one of the stock centers/NHPRR.**

rrids.org). The number of biomedical journals indexed in PubMed is 5294 (date accessed Dec 28, 2023; https://www.ncbi.nlm.nih.gov/nlmcatalog?term=currentlyindexed). Thus, even if we consider the larger number of 1000 journals, the probability of being asked to add RRIDs should be about 20%, and simply being asked to provide RRIDs does not result in compliance most of the time (Bandrowski et al, 2015 [4]). Furthermore, RRIDs did not enter journal publishing at the same time; 25 journals joined the initiative in 2014, but Cell Press journals joined in 2016–2018, while Nature and Science joined in 2018 and 2019. Journals continue to change their instructions to authors (see RRIDs.org). Authors may therefore not encounter the request for RRIDs, especially if they publish in journals that do not routinely ask for RRIDs. It is therefore highly important for compliance that stock centers also ask authors to cite stocks via RRID. From the data that we have gathered, it is a little difficult to determine which practices are most closely related to high compliance. MMRRC improved compliance within two years of implementing their new webpages, but reducing non-findable mice happened several years later, and a persistent campaign was led by the team for most of this time. AGSC drove the use of RRIDs and a reduction in the nicknamed animals, directly replacing the nicknaming practice with RRIDs. We should note that catalog numbers were not used in this community before the RRID project, and there is currently no genetic nomenclature authority for salamanders, so the community is quite centered on this stock center and is quite small. For NXR, all papers cited frogs using nicknames, but within two years, the percentage of nicknamed animals dropped precipitously, giving way to RRIDs and some more formal names. Although ZIRC did not have sustained campaigns for RRIDs, the community has been working with the ZFIN resource, which has been using ZDB numbers (which are also RRIDs for fish) since 1996, thus, there is already significant usage of PIDs in this community. The main reason that we don't believe that journals alone can change resource citation practices is because the NHPRR use case shows that even in places where RRIDs are well accepted, i.e., most journals that enforce RRIDs do so primarily for antibodies, there were relatively few RRIDs for NHPRR between 2014 and 2020. In 2021/2 the percentage of RRIDs rises quickly, though the trend toward findable resources is not significant. Thus, it seems that most of the time, the stock center website or campaign is the key to gaining compliance in a community of users, though journals can certainly play a part in getting some authors to comply with the standard.

### Sharing animals between labs: a word of caution

Our current method of reading papers does not address a very important and potentially problematic issue, which is resource sharing between labs, while still maintaining that the animals originally came from a stock center. Many authors state that the stock came from a stock center and was maintained in the university organism facility over a specified number of generations, but in other cases, when labs maintain their own colonies and share resources the provenance of the animal (full name or stock number) can be lost and instead the colonies can be referenced by a nickname. Long-term maintenance of a colony can also make it likely that there is substantial genetic drift, making data less comparable.

At the onset of this study, we attempted to ask authors in the "nickname" category which animals were used but found this to be far too labor-intensive for our study. In that category, very few papers specified how many generations their animals were kept in animal facilities after purchase, however, one researcher who was contacted had a colony of transgenic mice that originated from MMRRC. With the help of the author, we tracked down a sale of a single set of founders that was purchased by a nearby lab over 10 years before the paper was published. The researcher obtained mice from a colleague and brought the mice to a different university and did not re-derive the colony until our conversation (personal communication). This was a very clear case where an animal colony was established without taking into account potential genetic drift. Although this was abandoned as a systematic method for the current study, of the 40 authors contacted, about half of the authors were happy to provide the information, and some realized that the information should always be included in their manuscripts because it was difficult for them to track down. The other half of the authors were either non-responsive or let us know that the person who knew that information was no longer in the lab, and the information was therefore lost.

We did not attempt to examine papers in which there is no mention of a stock center, but most often these papers were far greater than the papers that did mention a stock center, for example there are ~400,000 zebrafish papers, but only 527 that mention the major, US national stock center over the same period (2011–2022). This may have multiple causes: 1. The stock center was simply omitted from the paper, 2. The stock was wild-caught or bought from a pet store, 3. The stock was obtained from another lab, and the provenance of the stock origin was lost between the labs, 4. The stock was generated de novo in the lab and was not deposited in a repository. In this last case, the wild-type stock likely came from a repository, but the information was omitted. While it is very likely that many labs have the in-house expertise to run a colony and do appropriate genetics controls, it is far more difficult to determine animal genetics, general animal health, and even accurate animal nomenclature in cases where the lab generates the animal or an animal is shared between labs. As journals begin to more frequently encourage authors to use RRIDs and proper nomenclature, the authors that generated these animals can register them with the nomenclature authority, e.g., MGI for mice or deposit them in a stock center, e.g., MMRRC.

We have seen the impact of RRIDs on the registration of research resources with plasmids. Addgene, which provides plasmids to the research community, until 2019 spent significant time soliciting plasmids from authors with limited success. Once several of the large journals requested that authors add RRIDs for plasmids and deposit them with Addgene, the project stopped needing to solicit submissions and is now dealing with a deluge of reagents (Personal Communication). Some of the submitted reagents do not make it through quality control at Addgene and are subsequently not made available to the community. Similar efforts to deposit and run quality control of all research resources may not be feasible or may be cost prohibitive, but identification of the genotype of a newly created organism with the genomic authority for that organism and the inclusion of quality control data for all newly created organisms should be the standard when publishing about a newly created resource.

## The cost of poor citation practices is astronomical

If we assume that the Freedman and colleagues (2017) [12] estimate that 50% of research is not reproducible and that the "non-findable" resources are the largest culprit (resulting in ~$10.8 billion in waste per year by their estimates) then then taking even relatively simple steps to improve findability of resources should lead to a more reproducible literature and less funding waste. We estimate that RRIDs when implemented by journals and stock centers increase findability by ~30% (see also Bandrowski et al, 2015 [4]); scaling up would suggest converting ~$3 billion in research funding from the non-reproducible to the reproducible category, at least if we consider that the only reason why a study is not reproducible is the lack of identification of resources. The stock centers, because of their central role as authorities for needed resources and outsized role at the center of their communities, can contribute substantially to improving how those communities keep track of animal and other resource provenance and report about this in the literature. Although the impact might be expected to be smaller than that provided by the stock centers, journals can also play a meaningful role in increasing the use of RRIDs, increasing rigor and reproducibility, and reducing waste.

## Resource citation metric

The citation of published papers is tracked by several indexing efforts, including the Web of Science, PubMed, and Google Scholar. Although the citation information rarely agrees across these, they are reliable enough that authors do not have to text-mine the scientific literature to determine the impact of their work. However, because research resources are not usually cited transparently or are cited in a way that is opaque to citation indexes, there is no easy answer to how many papers used a mouse or an antibody. Tracking the impact of research resources is critical for the continued support of university-based resource repositories because they are largely supported by grant funding. The staff at NHPRR has spent hundreds of hours tracking down citations to their reagents, time that can be seen as wasted effort. The RRID system of resource tracking is the closest thing to an index, such as the Web of Science, because each RRID is extracted from the scientific literature semi-automatically or automatically. Unfortunately, the system is only able to pick up at best

about 50% of the citations to mice and antibodies, most of the time the real number is closer to 10% of total due to current citation practices, limiting the effectiveness of this index.

Tracking down an RRID to assign a citing paper takes about 1 second in the fully automated case (when manuscripts are open access and under a text-mining accessible license) or about 10 seconds (when papers are not text-mining accessible, each paper takes about a minute to open and scan with an automated tool and there are about 10 resources per paper). Therefore, the cost of capturing citations for this manuscript was about 2 hours of FTE or about $200. For catalog numbers and easy lookups, we have estimated about 1 minute per citation, which translates to 25 hours of effort and about $2,500. The nicknames and the 'other' category took about 3 minutes to look up and took about 45 hours ($4,500) but provided very low-quality information or no information at all. When considering that there are about 274,000 papers in PubMed Central that referenced mice or antibodies in 2022, one can surmise that even spending 1 minute per citation makes it relatively cost-prohibitive to capture the resource used, as that would entail 2,740,000 minutes or about $4.5M in personnel time per year. These costs are borne by stock centers and other facilities, each spending significant resources to gather overlapping data, which are not verified by an objective third party.

The RRID system improves this calculation, making all data harvested by the RRID automated and semi-automated pipelines available on the specific RRID webpage, letting scientists know which paper used a particular antibody, but also letting the stock center know which salamanders are more highly cited. If the RRID system of citing resources is adhered to more broadly, the list of citations to each resource can indeed grow into a new resource impact index that will be useful in answering questions about resources.

## Limitations of this study

The data collected for NHPRR were collected very thoroughly but less systematically than we would have liked, making comparisons across stock centers a bit more difficult. The stock centers themselves are quite different, as are their user communities, with relatively little overlap. Thus, the author groups that are represented should be distinct, and their animal citation practices may be governed by the norms of those specific communities. This may be a factor with AGSC because it appears that most of those papers cite the name of the stock, usually inscrutable to biologists outside of the community, and in recent years, the RRID. In contrast, the mouse and zebrafish communities have been using stock center catalog numbers and model organism-derived persistent identifiers for at least 10 years, potentially making those communities pre-socialized to the concept of RRIDs. Another limitation of this study is that we did not have exact dates when some changes in websites and when campaigns were started at several stock centers, so we have used a coarser measure of the year when a change was implemented, as those were more certain. Another limitation is the lack of a control stock center which never in any way joined the RRID initiative, which would have made for a cleaner study, however our funding was specifically targeted towards the NIH Office of the Director at the supported stock centers, all of which were part of the RRID initiative in some way.

## Supporting information

**S1 File. Raw curation data underlying all figures is attached as a single file entitled** .
(XLSX)

## Acknowledgments

We would like to thank Mr. Nathan Anderson, who spent tireless days curating these data.

## Author contributions

**Conceptualization:** Martijn Roelandse, K. C. Kent Lloyd, Ian Korf, S. Randal Voss, Diogo M. Magnani, Zoltan Varga, Christina James-Zorn, Marko Horb, Jeffery S. Grethe, Anita Bandrowski.

**Data curation:** Agata Piekniewska, Giovanni de Castro, Anita Bandrowski.

**Formal analysis:** Agata Piekniewska.

**Funding acquisition:** Anita Bandrowski.

**Project administration:** Anita Bandrowski.

**Supervision:** Anita Bandrowski.

**Writing – original draft:** Agata Piekniewska, Anita Bandrowski.

**Writing – review & editing:** Agata Piekniewska, Martijn Roelandse, K. C. Kent Lloyd, Ian Korf, S. Randal Voss, Giovanni de Castro, Diogo M. Magnani, Zoltan Varga, Christina James-Zorn, Marko Horb, Jeffery S. Grethe, Anita Bandrowski.

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
