## [Decision Letter · Decision Letter 0]

PONE-D-25-07468Do organisms need an impact factor? Citations of key biological resources including model organisms reveal usage patterns and impact.PLOS ONE

Dear Dr. Bandrowski,

Thank you for submitting your manuscript to PLOS ONE. After careful consideration, we feel that it has merit but does not fully meet PLOS ONE’s publication criteria as it currently stands. Therefore, we invite you to submit a revised version of the manuscript that addresses the points raised during the review process.

Comments from Editor

Introduction

Please state clearly the authors who are attached to/associated with SciCrunch.

Please note that the Editor acknowledges the clear CoI statements provided elsewhere in the manuscript.

Methods

For completion, please comment on how retracted papers were dealt with.

Results

Please briefly explain what "Citations" versus "Papers" means, in Table 2; i.e., for MMRRC (row 1), 676 citations to MMRRC was made from 511 papers - is this correct?

For Figure 2 - please clarify how this data was generated; i.e., is the denominator from the row "Total - animal/antibody was used" in Table 2?

Figure 4

In the case of resources other than MMRRC (thus NXR, AGSC, ZIRC), the authors are asked to comment on whether the variation from linear could be due to the smaller sample sizes.

We look forward to receiving your revised manuscript.

Kind regards,

Miriam A. Hickey, PhD

Academic Editor

PLOS ONE

Reviewers' comments:

Reviewer's Responses to Questions

**Comments to the Author**

1. Is the manuscript technically sound, and do the data support the conclusions?

Reviewer #1: Yes

2. Has the statistical analysis been performed appropriately and rigorously? 

Reviewer #1: Yes

3. Have the authors made all data underlying the findings in their manuscript fully available?

Reviewer #1: Yes

4. Is the manuscript presented in an intelligible fashion and written in standard English?

Reviewer #1: Yes

5. Review Comments to the Author

Reviewer #1: The article from Piekniewska, et al examines the role of resource centers in using and encouraging the use of RRIDs by authors employing resources (animals and antibodies mostly) obtained from such centers. A major argument in the article, supported by time-intensive literature analyses, is that tracking of each resource is made far easier (and thus cheaper) when authors use RRIDs as algorithms designed to scan all available references can very quickly identify RRIDs. This is important for such stock centers, who are often heavily supported by NIH funding and thus better allows them to quantify use of their resources and impact in the literature. Another important aspect of the use of RRIDs, and one which I think is undersold in this article, is the value of using RRIDs (and thereby unambiguously identifying both the source and the specifics of the resource) to the rigor and reproducibility of the published works. The data support the increased use of RRIDs and that both publishers and resource centers are currently the best way to encourage or require their use. Although perhaps not the focus of this study, it seems appropriate to include specific conclusions regarding the added value of increased use of RRIDs to rigor and reproducibility. This is an early look at the impact of use of RRIDs and yet important trends are clearly emerging and supported by detailed statistical analyses. It is expected that this will be the first in a series of articles on this topic so will become a hallmark of sorts of its own. The use of RRIDs is an important topic that needs to be known about and understood by a larger fraction of the research community so publication is strongly supported. The following are suggestions, all minor, as to ways the manuscript might be improved.

The article needs a thorough and careful job editing for typos and wording. Just a few examples include “finable” instead of “findable” (twice), the word “data” is always plural (despite common use to the contrary), in the Abstract “heavily rely largely”.

I think this sentence should be clarified: “Unfortunately, the system is only able to pick up at best about 50% of the citations to mice and antibodies but most of the time the real number is closer to 10% of total due to current citation practices, limiting the effectiveness of this index.” There are other examples that hopefully will be identified and clarified in the detailed editing I encourage.

The use of SciScore is key to the analyses but I fear is not well known by most readers so including perhaps more details in the Methods section of what SciScore does and how it does it will both raise awareness of this great tool and also help readers understand its use in this manuscript and the data coming from it.

In a few places the authors seem to argue towards a specific conclusion but then fall short of simply stating their preferred outcome. One example, is the first paragraph under Sharing Animals Between Labs, a Word of Caution. Why not end with your point, e.g., “Best practice would argue in favor of including RRIDs for the original source but including a qualifier that includes the potential for genetic drift.” Other examples of stating the value of RRIDs not only for tracking use of resources by stock centers but also to help researchers find them and resulting in increased rigor and reproducibility.

6. PLOS authors have the option to publish the peer review history of their article (what does this mean? ). If published, this will include your full peer review and any attached files.

**Do you want your identity to be public for this peer review?** For information about this choice, including consent withdrawal, please see our Privacy Policy .

Reviewer #1: **Yes: ** Richard A Kahn

---

## [Author Response · Author response to Decision Letter 1]

21 May 2025

Dear Editors and Reviewers,

We would like to express our gratitude to the readers of our manuscript who suggested ways to clarify and strengthen the work. We acknowledge that there were some grammatical errors (data plural vs singular etc) that we had made in the original manuscript and have now gone through the manuscript using Grammarly to highlight many of these issues and we fixed those.

We have updated the manuscript and have uploaded the new files, including a version with changes as well as a manuscript version that is clean. We hope that this is now acceptable for publication in PLoS.

As to specific comments, please find them addressed below.

Author affiliations:

All authors who have a SciCrunch affiliation, are clearly listed as having a SciCrunch affiliation. In addition, the conflict of interest statement is included for each SciCrunch affiliated author. One author, Dr. Grethe, is a co-founder of SciCrunch Inc, but holds no position in SciCrunch therefore his affiliation remains only with UCSD, but since he holds significant shares he does have a COI statement. The repositories listed are all not-for-profit so authors affiliated with these repositories are not conflicted.

Retractions:

Thank you for drawing our attention to this key issue. We added a statement in the methods section specifying that: “We did not explicitly exclude retracted papers, but at the time of data collection curators did not find any papers that were retracted.”

Please briefly explain what "Citations" versus "Papers" means, in Table 2; i.e., for MMRRC (row 1), 676 citations to MMRRC was made from 511 papers - is this correct

We thank the reviewer for highlighting this key point that might bring confusion about citations of resources vs papers. Indeed it is correct that 511 papers cited 676 mice. A study typically uses ~10 RRIDs, often multiple mice (at least control and experimental groups which are frequently genetically distinct and have distinct RRIDs). We have clarified this point in a few places in the manuscript, including Table 2.

For Figure 2 - please clarify how this data was generated; i.e., is the denominator from the row "Total - animal/antibody was used" in Table 2?

We thank the editor for pointing out another area of potential confusion. There is only a sum and not a denominator for any analysis. We have added some explanatory text into the legend of table 2, also transcribed below, and we have added the resource type (fish, mouse) into the header row for readability.

“All data are reported as granular citations to a particular mouse or antibody used in a paper (citations) and the number of unique papers that cite the resources (papers). Numbers are counts of resources or individual papers that contain at least one citation of that category, for example 12 antibodies from NHPRR were cited by catalog number in 10 total papers. At the bottom we provide the raw numbers of google scholar searches for these resources with the goal of showing roughly the percentage of papers that were curated for this study vs the number of potential papers that were reviewed. Please note, NHPRR papers curated vastly outnumber google scholar search results because the stock center staff track authors who buy antibodies and follow up with individual authors who may not have mentioned the name of the stock center in their manuscript.”

Figure 4

In the case of resources other than MMRRC (thus NXR, AGSC, ZIRC), the authors are asked to comment on whether the variation from linear could be due to the smaller sample sizes.

We have added the following text to the section describing Figure 4 in the text pointing out that the smaller stock centers with fewer papers per year do vary more than the larger stock center.

“Please note that for stock centers other than MMRRC, the variation in average yearly SciScore is larger due to the smaller number of associated papers.”

Although perhaps not the focus of this study, it seems appropriate to include specific conclusions regarding the added value of increased use of RRIDs to rigor and reproducibility. This is an early look at the impact of use of RRIDs and yet important trends are clearly emerging and supported by detailed statistical analyses. It is expected that this will be the first in a series of articles on this topic so will become a hallmark of sorts of its own. The use of RRIDs is an important topic that needs to be known about and understood by a larger fraction of the research community so publication is strongly supported. The following are suggestions, all minor, as to ways the manuscript might be improved.

We thank the reviewer in the vote of confidence and we have included new portions of the conclusions/discussion highlighting the role of RRIDs in the quest for improved rigor and reproducibility.

“This change in individual communities of practice centered around the stock center underscores that improvements to rigor and reproducibility practices, especially those that are “about the resources” can indeed be augmented with the help of the stock centers. For investigators, RRIDs primarily serve as a means to ensure that they, or their readers, will easily find the exact research resources used in a study (Bandrowski et al, 2015; Marcus et al, 2016; Bandrowski, 2022), a necessary though not sufficient piece of the reproducibility puzzle. Any help or nudge that authors receive from stock centers and journals simply acts to push more authors to include these bits of unambiguous information about their reagents in their papers.”

I think this sentence should be clarified: “Unfortunately, the system is only able to pick up at best about 50% of the citations to mice and antibodies but most of the time the real number is closer to 10% of total due to current citation practices, limiting the effectiveness of this index.” There are other examples that hopefully will be identified and clarified in the detailed editing I encourage.

We have rephrased this sentence to highlight the problems with relying on text mining to find citations to resources. Thank you for pointing out this inconsistency in our language.

“Unfortunately, because the system relies in part on automatic extraction of text, it is only able to “see” about 50% of the citations to mice and antibodies, due in part to licensing restrictions of the content and the citation practices (e.g., nicknaming mice). In most cases, especially with older literature that is licensed with more restrictions and citation practices that rely on nicknames, the percentage of the literature available is closer to 10%, limiting the effectiveness of this index.”

The use of SciScore is key to the analyses but I fear is not well known by most readers so including perhaps more details in the Methods section of what SciScore does and how it does it will both raise awareness of this great tool and also help readers understand its use in this manuscript and the data coming from it.

We thank the reviewer for pointing out that we did not adequately highlight the role of SciScore. We have expanded the methods section describing some of the features of the tool. Also pasted below:

“SciScore is a specialized methods review tool for scientific articles. It employs 55 algorithms to evaluate submitted methods sections against a wide range of rigor criteria known to impact the reproducibility of scientific research. In addition to this, SciScore identifies sentences containing key research resources—such as antibodies, organisms, cell lines, plasmids, and software tools—and assesses how uniquely identifiable these resources are based on the accompanying metadata, e.g., the presence of a catalog number. Using this analysis, SciScore generates a reproducibility score and a detailed report, which includes a rigor adherence table and a key resources table. The adherence of the text to each of the applicable rigor criteria, e.g., inclusion of a statement about investigator blinding, increases the score. The details of the scoring metrics are published (Menke et al, 2022).”

In a few places the authors seem to argue towards a specific conclusion but then fall short of simply stating their preferred outcome. One example, is the first paragraph under Sharing Animals Between Labs, a Word of Caution. Why not end with your point, e.g., “Best practice would argue in favor of including RRIDs for the original source but including a qualifier that includes the potential for genetic drift.” Other examples of stating the value of RRIDs not only for tracking use of resources by stock centers but also to help researchers find them and resulting in increased rigor and reproducibility.

Thank you for asking for boldness. This is important. We have made small changes in several places in the manuscript highlighting the importance of the RRID system and more specifically we have added the sentence suggested about best practices for preventing genetic drift into the discussion section.

---

## [Decision Letter · Decision Letter 1]

PONE-D-25-07468R1Do organisms need an impact factor? Citations of key biological resources including model organisms reveal usage patterns and impact.PLOS ONE

Dear Dr. Bandrowski,

Thank you for submitting your manuscript to PLOS ONE. After careful consideration, we feel that it has merit but does not fully meet PLOS ONE’s publication criteria as it currently stands. Therefore, we invite you to submit a revised version of the manuscript that addresses the points raised during the review process.

We look forward to receiving your revised manuscript.

Kind regards,

Miriam A. Hickey, PhD

Academic Editor

PLOS ONE

Journal Requirements:

Additional Editor Comments:

Please address the comments from Reviewer 1.

Reviewers' comments:

Reviewer's Responses to Questions

**Comments to the Author**

1. If the authors have adequately addressed your comments raised in a previous round of review and you feel that this manuscript is now acceptable for publication, you may indicate that here to bypass the “Comments to the Author” section, enter your conflict of interest statement in the “Confidential to Editor” section, and submit your "Accept" recommendation.

Reviewer #1: (No Response)

2. Is the manuscript technically sound, and do the data support the conclusions?

Reviewer #1: Yes

3. Has the statistical analysis been performed appropriately and rigorously? 

Reviewer #1: Yes

4. Have the authors made all data underlying the findings in their manuscript fully available?

Reviewer #1: Yes

5. Is the manuscript presented in an intelligible fashion and written in standard English?

Reviewer #1: Yes

6. Review Comments to the Author

Reviewer #1: The manuscript from Piekniewska, et al, “Do organisms need an impact factor? …” has been clearly improved with appropriate and helpful edits in response to the previous review. This article makes an important contribution to the discussion surrounding RRIDs, citation of resources used in publications, and the roles of stock centers and journals. I fully support publication with the following minor edits or suggestions for consideration, in the order that they appear in the current manuscript.

1) Add DSHB to your list of abbreviations, with the full name?

2) Abstract: Change “From this data, …” to “From these data, …”

3) Define RRID the first time it is used in the text (currently top of p. 5) and consider also doing so in the Abstract as it is such an important part of the article.

4) Last sentence before Methods section, “university-based resource providers OVER 10 years, spanning the period before and …”

5) Heading right at start of Methods should include acronym used repeatedly later as it is the first time used, thus: NIH Office of the Director (OD) funded organism stock centers:

6) p. 10 under SciScore, define RTI the first time used (you do so later, but…).

7) Consider reversing the order of the two paragraphs under SciScore so that you first explain what SciScore is and then how you used it.

8) Figure 4 legend appears to have been cut off.

9) This sentence appears near the top of p. 22: “The main reason that we don’t believe that journals themselves can change resource citation practices is because the NHPRR use case shows that even in places where RRIDs are well accepted, i.e., most journals that enforce RRIDs do so primarily for antibodies, there were relatively few RRIDs for NHPRR between 2014 and 2020.” However, my understanding is that your data suggest that journals CAN have an impact, just that it is smaller than that made possible by the resource centers. Do you mean perhaps “journals alone” instead of “journals themselves”?

10) On p. 22 add of: “university organism facility for a specified number OF generations, but…”

11) In the next paragraph on p. 22, “over 10 years before the paper WAS PUBLISHED.”

12) p. 23, If we assume that THE Freedman and colleagues…”. In fact, please consider changing to something like the following” If we assume that the Freedman and colleagues (2017) estimate that 50% of research is not reproducible and that the “”non-findable” resources are the largest culprit (resulting in ~$10.8 billion in waste per year by their estimates) then …

13) Again, I am a bit uncertain as to the conclusions the authors are making as to the role of journals here. As I understand it they believe they can help but not as much as the resource centers, which are the focus of the article. Still, perhaps consider adding at the end of the paragraph at the bottom of p. 23 something like: Although the impact might be expected to be smaller than that provided by the stock centers, journals can also play a meaningful role in increasing the use of RRIDs, increasing rigor and reproducibility and reducing waste.

14) last sentence under Resource Citation Metric, you are missing a space: “antibodies,most”

15) p. 24, are these two sentences intended to be one?: “When considering that there are about 274,000 papers in PubMed Central that referenced mice or antibodies in 2022. One can surmise that even spending 1 minute per citation makes it relatively cost-prohibitive to capture the resource used, as that would entail 2,740,000 minutes or about $4.5M per year in personnel time per year.” And consider deleting one of the “per year”s.

7. PLOS authors have the option to publish the peer review history of their article (what does this mean? ). If published, this will include your full peer review and any attached files.

**Do you want your identity to be public for this peer review?** For information about this choice, including consent withdrawal, please see our Privacy Policy .

Reviewer #1: **Yes: ** Richard A. Kahn

---

## [Author Response · Author response to Decision Letter 2]

10 Jun 2025

We would again like to thank Dr. Khan for taking a thorough re-read and pointing out some additional omissions and small errors. We have addressed these in the revised document.

1) Add DSHB to your list of abbreviations, with the full name?

Please see page 1

“DSHB- Developmental Studies Hybridoma Bank (RRID:SCR_013527)”

2) Abstract: Change “From this data, …” to “From these data, …”

“From these data, it was not possible to determine whether outreach to authors or changes to stock center websites drove better citation practices, but findability of research resources and rigor adherence were improved.”

3) Define RRID the first time it is used in the text (currently top of p. 5) and consider also doing so in the Abstract as it is such an important part of the article.

Abstract: “The RRID (Research Resource Identifiers) project is filling this gap by working with journals and resource providers to improve citation practices and to track the usage of these key resources.”

Also on page 5: “The purpose of RRIDs (Research Resource Identifiers) is primarily to serve investigators to ensure that they, or their readers, will easily find the exact research resources used in a study (Bandrowski et al, 2015; Marcus et al, 2016; Bandrowski, 2022).”

4) Last sentence before Methods section, “university-based resource providers OVER 10 years, spanning the period before and …”

Sentence updated to: “We measured the citation practices of the research community for five university-based resource providers over 10 years spanning the period before and after RRIDs were introduced, and evaluated the ease with which automated routines could identify the organism or reagent used.“

5) Heading right at start of Methods should include acronym used repeatedly later as it is the first time used, thus: NIH Office of the Director (OD) funded organism stock centers:

Done: NIH Office of the Director (OD) funded organism stock centers:

6) p. 10 under SciScore, define RTI the first time used (you do so later, but…).

Updated sentence to:

“The curator collected the methods section from each paper and obtained the Reproducibility and Transparency Index RTI (Menke et al, 2020) for each paper using the SciScore tool (version 2, https://sciscore.com, RRID:SCR_016251).”

7) Consider reversing the order of the two paragraphs under SciScore so that you first explain what SciScore is and then how you used it.

Done; updated

8) Figure 4 legend appears to have been cut off.

Thank you, it has been recovered.

9) This sentence appears near the top of p. 22: “The main reason that we don’t believe that journals themselves can change resource citation practices is because the NHPRR use case shows that even in places where RRIDs are well accepted, i.e., most journals that enforce RRIDs do so primarily for antibodies, there were relatively few RRIDs for NHPRR between 2014 and 2020.” However, my understanding is that your data suggest that journals CAN have an impact, just that it is smaller than that made possible by the resource centers. Do you mean perhaps “journals alone” instead of “journals themselves”?

“Journals alone” is a better wording for this sentence.

10) On p. 22 add of: “university organism facility for a specified number OF generations, but…”

Sentence updated to:

“Many authors state that the stock came from a stock center and was maintained in the university organism facility over a specified number generations, but in other cases, when labs maintain their own colonies and share resources the provenance of the animal (full name or stock number) can be lost and instead the colonies can be referenced by a nickname.”

The maintenance of the stock in a university facility should be over a specified period of time / generations.

11) In the next paragraph on p. 22, “over 10 years before the paper WAS PUBLISHED.”

Updated sentence: “With the help of the author, we tracked down a sale of a single set of founders that was purchased by a nearby lab over 10 years before the paper was published.”

12) p. 23, If we assume that THE Freedman and colleagues…”. In fact, please consider changing to something like the following” If we assume that the Freedman and colleagues (2017) estimate that 50% of research is not reproducible and that the “”non-findable” resources are the largest culprit (resulting in ~$10.8 billion in waste per year by their estimates) then …

Updated to “If we assume that the Freedman and colleagues (2017) estimate that 50% of research is not reproducible and that the ”non-findable” resources are the largest culprit…”

13) Again, I am a bit uncertain as to the conclusions the authors are making as to the role of journals here. As I understand it they believe they can help but not as much as the resource centers, which are the focus of the article. Still, perhaps consider adding at the end of the paragraph at the bottom of p. 23 something like: Although the impact might be expected to be smaller than that provided by the stock centers, journals can also play a meaningful role in increasing the use of RRIDs, increasing rigor and reproducibility and reducing waste.

This sentence fits nicely and was added to the paragraph. We should certainly not let the journals off the hook here.

14) last sentence under Resource Citation Metric, you are missing a space: “antibodies,most”

Added thank you

15) p. 24, are these two sentences intended to be one?: “When considering that there are about 274,000 papers in PubMed Central that referenced mice or antibodies in 2022. One can surmise that even spending 1 minute per citation makes it relatively cost-prohibitive to capture the resource used, as that would entail 2,740,000 minutes or about $4.5M per year in personnel time per year.” And consider deleting one of the “per year”s.

Done thank you.

---

## [Editor Report · Decision Letter 2]

Do organisms need an impact factor? Citations of key biological resources including model organisms reveal usage patterns and impact.

PONE-D-25-07468R2

Dear Dr. Bandrowski,

We’re pleased to inform you that your manuscript has been judged scientifically suitable for publication and will be formally accepted for publication once it meets all outstanding technical requirements.

Kind regards,

Miriam A. Hickey, PhD

Academic Editor

PLOS ONE

Additional Editor Comments:

All comments have now been addressed satisfactorily.

Reviewers' comments:

None applicable.

---

## [Editor Report · Acceptance letter]

PONE-D-25-07468R2

PLOS ONE

Dear Dr. Bandrowski,

I'm pleased to inform you that your manuscript has been deemed suitable for publication in PLOS ONE. Congratulations! Your manuscript is now being handed over to our production team.

Kind regards,

on behalf of

Dr. Miriam A. Hickey

Academic Editor

PLOS ONE